# Advanced G-MPS-PMMA Bone Cements: Influence of Graphene Silanisation on Fatigue Performance, Thermal Properties and Biocompatibility

**DOI:** 10.3390/nano11010139

**Published:** 2021-01-08

**Authors:** Eva Paz, Yolanda Ballesteros, Juana Abenojar, Nicholas Dunne, Juan C. del Real

**Affiliations:** 1Institute for Research in Technology, ICAI, Comillas Pontifical University, Santa Cruz de Marcenado, 26, 28015 Madrid, Spain; yballesteros@comillas.edu (Y.B.); delreal@comillas.edu (J.C.d.R.); 2Mechanical Engineering Department, ICAI, Comillas Pontifical University, Alberto Aguilera 25, 28015 Madrid, Spain; 3In-Service Material Performance Group, Materials Science and Engineering and Chemical Engineering Department, “Álvaro Alonso Barba” Institute of Chemistry and Materials Technology, Universidad Carlos III de Madrid. Av. Universidad, 30, 28911 Leganés, Spain; abenojar@ing.uc3m.es; 4Centre for Medical Engineering Research, School of Mechanical and Manufacturing Engineering, Dublin City University, Stokes Building, Collins Avenue, D09 E432 Dublin 9, Ireland; 5School of Mechanical and Manufacturing Engineering, Dublin City University, D09 E432 Dublin 9, Ireland; 6School of Pharmacy, Queen’s University of Belfast, 97 Lisburn Road, Belfast BT9 7BL, UK; 7Department of Mechanical and Manufacturing Engineering, School of Engineering, Trinity College Dublin, D02 PN40 Dublin 2, Ireland; 8Advanced Manufacturing Research Centre (I-Form), School of Mechanical and Manufacturing Engineering, Dublin City University, Glasnevin, D09 E432 Dublin 9, Ireland; 9Advanced Materials and Bioengineering Research Centre (AMBER), Trinity College Dublin, D02 PN40 Dublin 2, Ireland; 10Advanced Processing Technology Research Centre, Dublin City University, D09 E432 Dublin 9, Ireland; 11Trinity Centre for Biomedical Engineering, Trinity Biomedical Sciences Institute, Trinity College Dublin, D02 PN40 Dublin 2, Ireland

**Keywords:** graphene, silane, bone cement, fatigue, biocompatibility, silanisation, thermal properties, mechanical properties

## Abstract

The incorporation of well-dispersed graphene (G) powder to polymethyl methacrylate (PMMA) bone cement has been demonstrated as a promising solution to improving its mechanical performance. However, two crucial aspects limit the effectiveness of G as a reinforcing agent: (1) the poor dispersion and (2) the lack of strong interfacial bonds between G and the matrix of the bone cement. This work reports a successful functionalisation route to promote the homogenous dispersion of G via silanisation using 3-methacryloxypropyltrimethoxy silane (MPS). Furthermore, the effects of the silanisation on the mechanical, thermal and biocompatibility properties of bone cements are presented. In comparison with unsilanised G, the incorporation of silanised G (G_MPS1 and G_MPS2) increased the bending strength by 17%, bending modulus by 15% and deflection at failure by 17%. The most impressive results were obtained for the mechanical properties under fatigue loading, where the incorporation of G_MPS doubled the Fatigue Performance Index (I) value of unsilanised G-bone cement—meaning a 900% increase over the I value of the cement without G. Additionally, to ensure that the silanisation did not have a negative influence on other fundamental properties of bone cement, it was demonstrated that the thermal properties and biocompatibility were not negatively impacted—allowing its potential clinical progression.

## 1. Introduction

In previous studies, it was demonstrated that the incorporation of well-dispersed Graphene (G) and Graphene oxide (GO) powder can be a promising solution in augmenting the mechanical performance of PMMA bone cement in an attempt to enhance the long-term survival of the cemented orthopaedic implants [1,2,3].

It has been demonstrated that the presence of G and GO nanoparticles within a polymer matrix produces a deviation and detention of crack fronts during their propagation, increasing the required energy for failure [4,5,6]. This reinforcing effect of G and GO has attracted considerable interest during recent decades, giving rise to advanced nanocomposites with enhanced mechanical properties [7,8]. In comparison with graphite, the use of these nanosized carbon derivatives presents very interesting advantages in the biomedical field, particularly in orthopeadic applications: e.g., the carbon derivatives are very efficient as reinforcements due to their high specific-surface area and show some biological properties such as antimicrobial activity and osteoconductivity [9].

Two crucial aspects determine the effectiveness of the G and GO as reinforcing agents: one is the homogenous dispersion of the nanoparticles and the other is the existence of strong interfacial bonds between the filler and the polymeric matrix. Different methods have been investigated to reduce the tendency of the G and GO nanoparticles to agglomerate and to promote the integration of these nanoparticles within the polymer matrix. However, obtaining stable dispersions of nanoparticles within the matrix is especially critical towards the large-scale production of nanocomposites, where most of the dispersion procedures conducted in the laboratory are not scalable for large-scale production [10].

One of the most promising solutions, that significantly improves the poor dispersion, helping to develop large-scale dispersion procedures of nanocomposites, is chemical functionalization [11,12]. The G and GO are easily functionalised thanks to their high reactive surface. A wide variety of functionalisation routes have been explored using different chemical compounds (e.g., epoxy groups, amines, isocyanate and carboxyl groups) [11] in seeking procedures that can allow easy and versatile customisation of the functional groups anchored to the surface of the nanoparticle. This allows adaptation of the surface chemistry of the nanoparticles to the chemical nature of the matrix, promoting through the functional groups the covalent bonding with the matrix and improving the dispersion within the matrix.

Graphene and its derivatives have also been extensively explored in the preparation of nanocomposites for biomedical applications, not only due to their reinforcement effect but also thanks to other interesting properties, such as their antimicrobial effect [13,14]. In the case of these nanocomposites for biomedical application, the functionalisation should comply with a particular requirement: the biocompatibility of the functionalised G and GO should be ensured [15,16].

It has been reported that the silanisation of carbon-based nanomaterials can be an effective solution to improve their dispersion during the preparation of nanocomposites [17,18,19,20]. In previous studies [21], a successful functionalisation route has been reported to promote the dispersion of G in the PMMA bone cement via silanisation using 3-methacryloxypropyltrimethoxy silane (MPS). The silane coupling agent acts as an interface between the inorganic substrate (i.e., G) and the organic matrix (i.e., the PMMA bone cement), thereby promoting the interfacial reactivity of two dissimilar materials. The MPS silane has demonstrated an adequate biocompatibility in several applications [22,23,24] allowing their potential application in the preparation of G-PMMA bone cements. 

Previous research [21] has demonstrated that PMMA bone cements prepared with 0.1 wt.% of silanised G (G-MPS) showed improved fracture toughness, bending and compression strength in comparison with the unfunctionalised G. However, it is still necessary to further investigate the potential of these G-MPS nanoparticles for the mechanical performance of the PMMA bone cement under fatigue loading, as well as the effect of this silanisation on other fundamental properties (i.e., the thermal properties and biocompatibility) to ensure their potential clinical application.

In an attempt to establish the advantage that the silanisation could provide in the preparation of G reinforced bone cement, this work aimed to study the fatigue behaviour, thermal properties (i.e., maximum temperature, setting time, curing heat, residual monomer and thermal conductivity) and biocompatibility of G-MPS-PMMA bone cements. Additionally, a further study investigating the performance of the cement under bending has been completed. PMMA bone cements with unfunctionalised G and without G (Control) will be also analysed for comparative purposes.

## 2. Materials and Methods

### 2.1. Nanomaterials

In this study, graphene (G) powder (Avanzare Nanotechnology, Navarrete, Spain) was chemically functionalised via silanisation and incorporated to a PMMA bone cement as reinforcement. According to the supplier data sheets, the G powder was composed of 1–2 layers of G sheets with an average lateral size of 50–500 nm and a thickness of 0.7 nm.

For the G silanisation, a silane coupling agent, 3-methacryloxypropyltrimethoxysilane (MPS) (ABCR Gmbh, Karlsruhe, Germany) was used. The detailed silanisation procedure has been previously reported [21]. The adopted oxidation and silanisation routes are based on the routes developed by other authors; this literature has been used as a starting point and some modifications have been introduced to optimise the procedure [17,19,20]. The silanisation comprises two steps: (1) oxidation of G surface; the presence of oxygenated groups is necessary to anchor the silane molecule to the G surface, and (2) silanisation via covalent reaction of silane molecules with the oxygenate groups present in the oxidised G surface.

The G oxidation was performed following two different routes to evaluate the effect of the G surface oxidation on the silanisation performance. (1) The first route was a two-stage procedure. Firstly, 0.3 g of G powder was mixed by magnetic stirring with 70 mL of HNO_3_ (3 M) at 60 °C for 15 min. Then, the suspension was transferred to covered glass tubes and placed in an ultrasonic bath (Elmasonic p60h, Elma Schmidbauer GmbH, Singen, Germany) for 1.5 h at 37 Hz and 100 W. During the second step, the resultant G was treated with 70 mL of H_2_O_2_ (30% *w/v*) following the same described procedure (i.e., stirring and sonication). The G oxidised by this route was named G-Oxi1. (2) The second route was a one-stage procedure in which 0.5 g of G powder were added to 100 mL of a solution of H_2_SO_4_ (96%): NHO_3_ (3 M) (75:25 *v/v*) at 60 °C; the suspension was magnetically stirred for 15 min at 800 rpm, then this was sonicated with an ultrasonic bath for 2 h at 37 Hz and 100 W. The G oxidised by this route was named G-Oxi2. In both routes the resultant oxidised G powder was washed, filtrated and freeze-dried prior to silanisation using a LyoQuest freeze dryer (Telstar, Madrid, Spain).

The silanisation of the oxidised G was performed following the same procedure in both cases, 100 mL solution of ethanol:deionised water (80:20 *v/v*) was prepared and the pH was adjusted to 3.5−4.5, then the same mass of MPS to the mass of nanoparticle to be silanised was added to the solution and was magnetically stirred for 30 min at 800 rpm under ambient conditions (silane hydrolysis). Thereafter, the oxidised G (G_Oxi1 or G_Oxi2) was added to the solution and was dispersed using ultrasonication (Digital Sonifier 450, Branson Ultrasonics Corporation, Danbury, CT, USA) for 10 min at 50% of amplitude. Consequently, the solution was magnetically stirred for 2 h at 800 rpm at 65 °C. Once the liquid evaporated, the residual powder was washed and freeze-dried. The dried powder was then placed into an oven held at 120 °C for 2 h to improve the crosslinking of the silane molecules. The silanised G obtained from G_Oxi1 and G_Oxi2 was denoted as G_MPS1 and G_MPS2.

### 2.2. PMMA Bone Cement

The PMMA bone cement used in this study was a two-phase bone cement. The solid phase was composed of 3.64 g of barium sulphate (Sigma Aldrich, Madrid, Spain) and 36.36 g of Colacryl B866 (Lucite International Ltd., Wilton, UK), which contained the pre-polymerised PMMA and initiator (benzoyl peroxide, BPO). The liquid phase was composed of 19.9 mL of the methyl methacrylate (MMA) and 160 µL of an activator, N,N-Dimethyl-p-toluidine (Sigma Aldrich, Madrid, Spain. The bone cement formulation described has been used in previous studies and is analogous to the commercial bone cement, DePuy CMW1 [1,25].

The PMMA bone cements without G (Control), with 0.1 wt.% of unfunctionalised G and with 0.1 wt.% of each of the two silanised powders (G-MPS1 and G-MPS2) were prepared. The G powder was dispersed into the liquid phase of the bone cement using ultrasonication at 50% amplitude for 3 min at intervals of 30 s ON and 10 s OFF. To prevent overheating, the liquid monomer was placed in a water bath that was held at 22 ± 1 °C. Following sonication, the suspension was placed in an ultrasonic bath for 1 min, to reduce the incidence of bubble formation.

### 2.3. PMMA Fatigue Testing

Fatigue properties of the cements were determined using half-sized ISO 527-5 multipurpose test specimens [26]. The dimensions for each fatigue specimen were length 75 ± 0.5 mm, width 5 ± 0.2 mm, thickness 3.5 ± 0.2 mm, with a gauge length of 25 ± 0.5 mm. Specimens were prepared by injecting the PMMA bone cement into silicone moulds with the specific dimensions. Cement was polymerised in the mould for 24 ± 1 h. Thereafter the specimens were removed from the mould and the rough edges were sanded using abrasive paper. Fatigue testing was conducted under room conditions (22 ± 1 °C) using an electrodynamic testing machine (ElectroPuls E3000, Instron, Norwood, MA, USA). Each specimen was tested in compression-tension until failure or until 1 million cycles, using a sinusoidal stress cycle with a maximum stress of 11 MPa and a stress ratio of R = −1 according to ASTM F2118-03 [27]. It has been reported that the stress levels in the bone cement mantle around a joint replacement range from 3 to 11 MPa [28]; consequently, a maximum stress of 11 MPa has been used in order to ensure the success of the prosthesis under the most adverse conditions. A frequency of 2 Hz has been used for the test following ASTM F2118-03, which recommends not to exceed 5 Hz to avoid frequency effects. A total of three batches were tested for each cement composition with a minimum of five samples per batch.

Fatigue test results were analysed using four different methodologies [29,30,31]: (1) Probability of fracture method [32,33], i.e., the 50% of fracture life *N*_50_ was obtained for each cement combination; (2) Three parameter Weibull method [32,33]: these parameters were the Weibull minimum fatigue life, *N*_0_ (least number of cycles before failure), the Weibull characteristics fatigue life, *N_a_* (representing the life at 36.8% survival of the population; higher values of *N_a_* indicate a higher mean fatigue strength) and the Weibull slope or shape parameter, *b* (denoting the extent of scatter associated with a particular specimen set); (3) Probability of survival method [34]. The curves of the survival probability (*P_s_*) versus the number of cycles to failure (*N_f_*) were plotted; and (4) Fatigue performance index, *I* [35], which is used to convey the concept that fatigue performance is dependent on the values of both *N_a_* and *b*, since a good fatigue performance requires both a long fatigue life (i.e., a high value of *N_a_*) and high predictability of *N_f_* results (i.e., a high value of *b*).

### 2.4. Bending Properties

A four-point bending load arrangement was used to determine bending properties in accordance with ISO 5833 [36]. Specimens were in the form of rectangular bars with dimensions of 80.0 ± 0.1 mm length, 10.0 ± 0.1 mm width and 4.0 ± 0.1 mm thickness. The tests were conducted using a Universal Testing Machine IBTH/500 (Ibertest, Madrid, Spain) using a load cell of 5 kN, which operated at a crosshead speed of 5 mm/min. A total of three batches were tested for each cement composition with a minimum of five samples per batch.

### 2.5. Microscopy Analysis

Atomic Force Microscopy (AFM) was used to analyse the effect of the silanisation on the morphology of the G powder using a Nanotec Cervantes AFM system controlled with WSxM software (Nanotec Electronica, Madrid, Spain) [37]. Tapping mode imaging was performed using cantilevers with resonance frequency of 150 kHz, force constant 5 N/m and tip radius less than 10 nm. For specimen preparation, the G powder was dispersed in MMA, following the same procedure described in the preparation of the cement, then the G-MMA solution was sprayed onto a mica disc, then the MMA was evaporated using N_2_. The mica disc with the deposited G was then introduced in the AFM for analysis. For each kind of G, three mica samples were prepared and for each sample at least four different spots were measured with the AFM.

Scanning Electron Microscopy (SEM) analysis of the fractured surfaces of the fatigue specimens was conducted using an XL-30 Scanning Electron Microscope (Philips, Eindhoven, Holland). The energy of the electron beam was 10 kV. Each specimen was gold coated using a high-resolution Polaron SC7610 sputter coater (VG Microtech, East Sussex, UK), which provided a conducting medium for the electrons and enough contrast for the SEM images.

### 2.6. Thermal Properties

#### 2.6.1. Maximum Temperature and Setting Time

The maximum temperature (T_max_) and setting time (t_set_) were determined in accordance with ISO 5833 [36]. The PMMA bone cement was contained within a PTFE mould and the progression of temperature during polymerisation was measured using a nickel/chromium/aluminium K-type thermocouple. The measurements were registered at 1 sec intervals for 60 min using data logger MV1000 (Yokogawa Electric Corporation, Tokyo, Japan). For each cement combination, the test was run in triplicate. The T_max_ was determined as the highest temperature registered. The t_set_ was denoted as the time from the beginning of mixing until the temperature of the polymerising mass reaches the setting temperature (T_set_), where T_set_ is defined as the mean value between T_max_ and T_amb_ (i.e., recorded ambient temperature).

#### 2.6.2. Curing Heat and Residual Monomer Content

Differential Scanning Calorimetry (DSC) was used to determine two parameters: (1) the heat released during the complete cement polymerisation (i.e., the curing heat) and (2) the unpolymerised monomer content following isothermal polymerisation of the cement (i.e., residual monomer content). Tests were conducted using a DSC822 (Mettler Toledo, Greifensee, Switzerland) and liquid nitrogen was used as the purge gas (80 mL/min). Approximately 5–10 mg of uncured bone cement was placed in an aluminium crucible of 40 µL with a hole in the lid. Each DSC test commenced at four min from the start of cement mixing. Each bone cement composition was tested in triplicate.

Two different tests were performed to determine each parameter. (1) Dynamic test was used to determine the curing heat. Each dynamic test was performed between 0 and 200 °C at a scan rate of 10 °C/min. The heat produced during the cement polymerisation was determined by calculating the area under the heat flow versus time plot; and (2) Isothermal test was used to determine the residual monomer. Each isothermal test was performed at 22 °C for 45 min. The energy released during polymerisation (Q_iso_) was determined as the area under the heat flow versus curing time plot. A second segment of dynamic scanning from the 22 to 200 °C, at a scan rate of 10 °C/min was also included in the isothermal test and the heat released was determined (Q_dyn_). The residual monomer content was calculated using Equation (1)—where Q_Total_ is the total polymerisation heat required to complete polymerisation (Q_iso_ + Q_dyn_).
Residual Monomer (%) = (Q_dyn_/Q_Total_) × 100(1)

#### 2.6.3. Glass Transition Temperature

Glass transition temperature (T_g_) was also measured using DSC with the polymerised bone cement samples. In this case, a scan from 0 to 200 °C at 20 °C/min was completed, and the T_g_ value was measured at the midpoint according to the DIN 51007 standard. DSC was carried out at 20 °C/min as it allows observation of small changes that would not be easy to find at lower heating rates.

### 2.7. Biocompatibility Studies

To study the extent of biocompatibility, disc-shaped samples of bone cement (thickness of 2.0 ± 0.1 mm and 12 ± 0.1 mm of diameter) were incubated with an osteoblast precursor cell line (MC3T3) derived from mouse calvaria. The level of MC3T3 viability was determined after 72 h in a culture medium using the CellTiter 96 aqueous cell proliferation assay (Promega Corporation, Fitchburg, MA, USA). Following an incubation period of 72 h, 20 µL of MTS reagent was added to each sample with 100 µL of culture medium. The sample was then incubated for 4 h at 37 °C in a humidified 5% CO_2_ atmosphere. The absorbance was recorded at 490 nm using a Universal Microplate Reader EL 800 (BioTek Instruments, Inc., Winooski, VT, USA). The absorbance values recorded were determined to be proportional to the number of viable cells proliferating on each cement surface. The correlation between the number of cells and the absorbance at 490 nm was estimated and was used to calculate the number of viable cells in each sample using the corresponding calibration curve. Six samples were tested for each cement type.

### 2.8. Statistical Analysis

Each property was expressed as mean ± standard deviation. The results were also evaluated for statistical significance using a one-way analysis of variance (ANOVA) test with a post-hoc Scheffe’s test (SPSS 20.0 for Windows; IBM SPSS, Chicago, IL, USA). A *p*-value less than 0.05 was indicative of statistical significance.

## 3. Results and Discussion

### 3.1. AFM Analysis

AFM analysis aims to determine if silanisation can reduce the tendency of G to form aggregates. For this purpose, the AFM samples were also prepared following the same dispersion method: the same ultrasonication time, amplitude and solution concentration. After sonication, a drop of the G-MMA solution was deposited on a mica disc and the MMA was evaporated. In this way, the AFM analysis of the G agglomerates observed over the mica surface can provide a measure of the extent of homogenous dispersion.

In order to achieve an improvement in G dispersion, it is crucial that, during the silanisation, the morphology of the G sheets do not suffer substantial modifications that could compromise their performance as reinforcement (e.g., breakdown of G sheets reducing their specific surface area or formation of agglomerates) [38]. Figure 1 shows some representative images of G sheets aggregated observed during the AFM analysis. In general, it was noted that the superficial dimension of these G sheets was not affected by the silanisation procedure, which means that the silanisation did not produce any breakage of G sheets.

The Z-profiles (Figure 1) provided information about the thickness of the G agglomerates and, therefore, the extent of achieved exfoliation. It was observed that in the case of the unsilanised G, the extent of exfoliation was relatively poor. The thickness of G clusters was approximately 100 nm (Figure 1a), taking into account that according to the G data sheet, each G layer is about 0.7 nm in thickness, which means that the agglomerates are formed by approximately 142 layers of G. However, in the case of silanised G (G_MPS1 and G_MPS2) (Figure 1b,c), a considerable reduction in the agglomerates formed was observed. The agglomerates of G_MPS1 showed a thickness of around 25–35 nm (i.e., 35–50 layers) and G_MPS2 around 35–50 nm (i.e., 50–70 layers). This confirms that silanisation favours the exfoliation of the G sheets and avoid the formation of aggregates. Similar results were also observed by other authors [39]. It is also notable that according to the AFM data, the extent of exfoliation is higher in the case of G_MPS1 than G_MPS2, which agrees with the higher degree of silanisation of G_MPS1 observed in previous studies [21]. This study was focused on the characterization of G through the silanisation process and demonstrated that the G oxidation by route 1 produces a higher amount of hydroxyl groups on the G surface. These hydroxyl groups promote the anchoring of MPS molecules, leading into a higher degree of silanisation than in the case of the route 2.

### 3.2. Bending Properties

It can be observed from the bending results (Table 1), that the presence of 0.1 wt.% G did not produce significant modifications in bone cement properties according to ANOVA analysis: i.e., the bending strength decreased by 1.9% (*p* = 0.997) and the bending modulus increased 5.7% (*p* = 0.902) when compared with the Control. However, when the same amount of silanised G was incorporated, interesting improvements in comparison with unsilanised G were observed. For both G_MPS1 and G_MPS2, the results are very similar when compared with the Control, e.g., the bending strength increased by 14%, bending modulus by 21% and deflection at break by 12%.

A representative bending test graph (i.e., Load vs. Deflection) for each bone cement type is shown in Figure 2. The incorporation of silanised G into the bone cement resulted in a high deflection at break with high bending strength and modulus, which suggests the toughness of the bone cement.

### 3.3. Fatigue Testing

The fatigue test data demonstrated that the G silanisation provides a marked improvement on the bone cements fatigue performance. Figure 3 shows that, in comparison with Control cement, the incorporation of unsilanised G increased the mean number of cycles to failure by 524% (*p* = 0.423). However, this improvement is by 974% (*p* < 0.01) for G_MPS1 and by 743% (*p* = 0.187) for G_MPS2. This means that, in comparison with the use of unsilanised G, the silanisation in the case of G_MPS1 increased the mean number of cycles to failure by 186% (*p* = 0.665) and of G_MPS2 by 142% (*p* = 0.974).

It is noted that the enhancement caused by silanisation is more pronounced for the G_MP1 when compared to the G_MPS2. This could be a consequence of the higher level of homogenous dispersion achieved in the case of G_MPS1, which was observed from the AFM analysis and can be attributed to a higher silanisation degree.

Additionally, in the case of unsilanised G it is important to mention that the fatigue tests demonstrated notable improvements, unlike the bending tests, where no differences with respect to the control were obtained. This can be attributed to the fact that the mechanism by which the G improves the mechanical performance of composites (crack arrest and deviation) has a higher impact over the failure mechanisms produced during fatigue failure that during static tests [6].

Although the analysis of the main cycles to failure showed the effectiveness of the silanisation in terms of fatigue performance, this parameter typically has a high level of data scattering (as indicated by the large error bars in Figure 3), it being difficult to obtain significant differences from a statistical point of view. Therefore, a Weibull analysis was done to obtain more robust results and indicators. The estimators of the fatigue life are summarised in the Table 2 along with the rate of variation of each parameter when is compared with the Control group and the unsilanised G.

Weibull analysis showed that incorporating 0.1 wt.% of unsilanised G to the Control bone cement increased by more than 5 times (×5) the number of cycles at which the 50% of the specimens failed (*N*_50_) and the Weibull characteristic life (*N_a_*) (Table 2). These improvements are notable greater in the case of G_MPS1 and G_MPS2, demonstrating the significant effect of silanisation. The same parameters, *N*_50_ and *N_a_* in comparison with the Control, increased by twelve times (×12) in the case of G_MPS1, and by ten times (×10) for G_MPS2, which meaning that silanisation duplicate the values obtained by unsilanised G. In a similar way to the main number of cycles, it can be noted that the results are better in the case of G_MPS1 than G_MPS2.

The parameter *b* is an indication of the extent of data variance. It can be noted that this parameter was reduced in all cases in comparison with the Control cement. However, this reduction was lower for G_MPS1 and G_MPS2 when compared to the G, indicating that although silanisation increased fatigue parameters, the variability of the results was higher.

The fatigue performance index (*I*) is an interesting parameter, which evaluates at the same time the improvement and the variability of the fatigue results. This is a better global estimator of the fatigue performance since it considers both the *N_a_* and *b* parameters. The analysis of the *I* value corroborated the commented improvements: i.e., the presence of G greatly increased the fatigue index of Control cement (×4.3 times) and these improvements were doubled on silanisation of G: ×10.5 times in G_MPS1 and ×9.4 times in G_MPS2.

Applying the Weibull theory, the probability of survival (*P_s_*) for a given number of cycles was determined for all cements (Figure 4). For example, for a given *P_s_* of 0.3, the Weibull life for the Control specimens was 22,000 cycles, i.e., 30% of the Control cement survival beyond 22,000 cycles. In contrast, incorporating G into the bone cement increased the Weibull life to 128,000 cycles for the same *P_s_* level. The enhancement was even notably higher in the case of G_MPS1 and G_MPS2, achieving for the same level of *Ps*: 335,000 and 218,000 cycles.

It is important to comment that in comparison with fatigue data, the improvements obtained in bending properties were less relevant. These notable difference between the effect that G has over static mechanical properties (i.e., bending strength and bending modulus) and fatigue performance are typically reported in the literature [40,41]. This is attributed to the mechanism by which G improves the mechanical performance of materials, i.e., through crack deviation and arrest. This mechanism has a greater influence during the failure as a consequence of fatigue loading when compared to static loading, irrespective of the mode of state loading (i.e., bending, compression or tensile) [6,42].

Fatigue data demonstrated that the functionalisation of G via silanisation with MPS is a highly effective solution to obtain PMMA bone cements with superior mechanical performance. The enhancement of fatigue properties are especially important in such applications that suffer dynamical loads, as is the case of PMMA bone cements, constantly subject to the loads and stresses caused during the normal daily activity of the human body; it is well known that fatigue cracking of cement mantle is one of the main causes of prosthesis failure [43,44].

The positive impact of silanisation can be attributed to two fundamental aspects: (1) the presence of silane molecules bonded to the G surface favours the homogenous dispersion of G sheets, avoiding the restacking caused by the Van der Wall forces, and (2) the silane groups allow the creation of covalent bonds in the interfacial surface between G and PMMA acting as coupling agents. These produce stronger interfacial bonding between the PMMA bone cement matrix and the G surface, achieving superior mechanical properties by improving the interfacial load transfer. Some studies [45,46] demonstrated that this improvement in fatigue performance is due to the formation of stronger interfacial interactions that lead to an increase in energy absorbed caused by a transition in crack propagation mechanism (i.e., from interfacial slippage to crack arresting behaviour). Some simulations revealed that such a change in mechanism is caused by the formation of both hydrogen bond networks and physical entanglements at the interface [45].

### 3.4. Fractographic Analysis

The fractured surfaces of the fatigue specimens were evaluated using SEM in order to understand the mechanism by which the addition of G modifies the fatigue performance of the bone cement.

The fractured surfaces of the Control cement exhibited characteristics typical of brittle failure (Figure 5), i.e., a smooth surface and glossy aspect. Figure 5c shows multiple crack initiation sites along the surface with the presence of fatigue striations or rib markings on the fracture surface (these striations are highlighted in Figure 5c).

The white agglomerates indicated (by arrows) in Figure 5b,c denote the radiopaque agent (barium sulphate) used in PMMA bone cement). The barium sulphate was observed in a similar distribution and proportion for all fractured surfaces studies, irrespective of G. 

When the fracture surfaces of unsilanised G cements were studied, although some regions demonstrated homogenous dispersion of the G within the PMMA matrix many other areas showed poor dispersion or a lack of integration of the G sheets within the matrix. For example, in Figure 6a it can be observed some G sheets integrated well within the matrix (indicated by arrows). However, some examples of poorly dispersed G (indicated by arrows) can also be observed in Figure 6b,c. In Figure 6c, the highlighted region indicates a defect caused by the presence of poorly dispersed G, where the lack of adhesion between the G and the PMMA can be appreciated.

On the contrary, when the fractured surfaces of fatigue specimens with G_MPS1 and G_MPS2 were analysed, a better interaction between the G sheets and the PMMA matrix was observed. In Figure 7, a better degree of integration between the G and the PMMA matrix is observed. The location of the G position is again indicated in the image by arrows. The gap between G_MPS1 and G_MPS2 sheets and the PMMA matrix is significantly less when compared to the unsilanised G (Figure 6c). These results demonstrated that the silanisation produced a notable enhancement in G dispersion and interaction with the PMMA matrix.

The bone cements containing silanised G demonstrated better adhesion between G sheets and the polymer matrix, corroborating that stronger interfacial bonds have been created, promoting and facilitating the mechanism by which the presence of G improves the cement’s performance under fatigue loading. Several mechanisms by which the presence of G improves fracture energy and fatigue life of composites [6,47] have been previously reported; among the most important are: (1) Pull-out: G pulls out the crack from the matrix and slows down crack propagation by the interfacial friction between graphene and matrix; (2) Crack deflection: crack deflects into a different plane when it encounters G, resulting in a tortuous path and more energy dissipation for crack propagation; (3) Crack tip shielding: the crack tip is restricted in the vicinity of G due to the insufficient energy required for interfacial debonding. It is important to note all these mechanisms are directly related to the extent of interfacial strength between G and the PMMA matrix.

### 3.5. Thermal Properties

Neither the incorporation of G into the PMMA matrix nor silanisation significantly altered the thermal properties of bone cement (Figure 8).

However, some minor effects were noted, in particular, it was interesting to comment that the addition of G_MPS1 and G_MPS2 seemed to slightly decrease the extent of the curing heat and the maximum temperature (*T_max_*) reached during the cement polymerisation. The curing heat decreased from 91.6 ± 13.1 (J/s) for the Control to 71.4 ± 18.8 (J/s) and 66.3 ± 18.4 (J/s) for the G_MPS1 and G_MPS2. The *T_max_* decreased from 73.4 ± 4.3 °C for the Control to 57.0 ± 4.7 °C in G_MPS1 and 61.9 ± 2.9 °C in G_MPS2.

The polymerisation reaction of PMMA bone cement is highly exothermic, resulting in temperatures in the cement mantle which can rise to 100 °C [48,49]. These elevated temperatures can cause significant damage in the bone cells by thermal necrosis and potential loosening of the prosthesis [50]. Consequently, any reduction in curing heat could be highly beneficial to prevent the failure of the prosthesis caused by thermal injury.

Other studies have reported a reduction in the reaction exotherm of PMMA bone cements as a consequence of the presence of carbon-based nanomaterials (e.g., carbon nanotubes, graphene and graphene oxide) [1,3,51]. This phenomenon has been attributed to a retardation in the polymerisation reaction as a consequence of the active role that these nanomaterials can have on the free radical polymerisation process, acting as radical scavengers during the radical reaction due to their high reactivity. Two different types of actions have been described that can restrict the polymerisation: the first one is the retardation by the reaction of nanomaterials with the primary radicals formed from the BPO molecules, giving rise to non-reactive species and inhibiting the formation of the free radicals which cause the polymerisation reaction of the MMA molecules. The second one is the inhibition of the polymer chain growth by reaction with the chain and ending the growth, resulting in the formation of small-sizes molecules [3,52].

Additionally, it also has been reported that the extent by which these mechanisms take place is directly related to the level of chemical interaction between the nanomaterial and the PMMA bone cement during the reaction [53]. Taken this into account, the obtained results suggest that the silanisation is favouring the described radical inhibition. Additionally, this retardation in the polymerisation can corroborate the fact that silane functional groups are reacting with the MMA during the bone cement curing, creating covalent bonds between G and PMMA that help to explain the observed stronger interfaces.

Furthermore, it has been reported that the retardation of the polymerisation leads, in turn, to an increase in the setting time and residual monomer. Both effects, contrary to the temperature decrease, could negatively affect the clinical application of the bone cement, depending on the degree of retardation. On the one hand, an increase in setting time could hamper the alignment and the stability of the prosthetic implant, and on the other hand, an increase in residual monomer could promote the chemical necrosis due to the toxicity of MMA monomer [54]. However, these effects were not observed in the obtained results with the G-MPS1 and G_MPS2, ensuring that the silanisation does not negatively affect other fundamental properties of bone cements.

Additionally, the obtained values of *T_g_* were similar for all bone cements (*p* = 1.00). These results confirm that the presence of G did not produce changes to the polymer chain length, which suggests that despite the observed polymerisation retardation, this did not have a negative impact on the polymer chain growth.

### 3.6. Biocompatibility Studies

The biocompatibility of the G reinforced bone cements is an important issue to ensure their potential use in vivo. The cytotoxicity was analysed by determining the viability of MC3-T3 cells when exposed to the different bone cement types as a function of the viable cell number following an incubation period of 72 h (Table 3). The cell viability of unsilanised G bone cements was reported in previous works where their biocompatibility was demonstrated [55]; in the current work the cell viability of silanised G was evaluated in order to ensure that the silanisation did not affect the biocompatibility. The data of unsilanised G were also included in Table 3 for comparative purposes.

The results indicate that neither the presence of G nor the silanisation invoked a cytotoxic response, which demonstrated an adequate level of biocompatibility. Although small reductions in the number of cells were obtained in the cements with G, no statistical differences (p > 0.05) were observed when the Control cement was compared to the G, G_MPS1 and G_MPS2 bone cements. These data support the results of other studies that demonstrated adequate biocompatibility of silanes agents [22,56,57].

Given the promising results obtained, future work will be completed to continue investigating the biological benefits and effects of the G_MPS_PMMA bone cements and also the potential drawbacks with a deeper biological study. These studies will include antimicrobial tests, since some studies has reported that G and also silanisation could promote antimicrobial activity [9,58], and cell adhesion and proliferation tests, since G and their derivatives have proved to promote bone cell growth and regeneration [9,59]. In addition, in the case that biological testing will show favourable results, it will be interesting to determine the optimum level of loading not only from a mechanical point of view, but also from a biological perspective. Finally, it will be necessary to evaluate the mechanical and biological performance of the optimal bone cement using small animal in vivo experiments.

## 4. Conclusions

The present study describes an effective method for G functionalisation via silanisation with MPS that markedly improves the static and fatigue mechanical properties of PMMA bone cement. Specifically, silanisation of G nanoparticles significantly increased the mechanical properties of PMMA bone cements filled with 0.1 wt.% of G, and in particular the fatigue performance. The fatigue tests demonstrated the fatigue performance index (I) of PMMA bone cements with G_MPS was 200% higher when compared to the I value obtained when unsilanised G was incorporated into the PMMA matrix and 900% higher than the Control cement. This great improvement is attributed to two fundamental mechanisms: (1) the presence of silane molecules bonded to the G surface favours homogenous dispersion of G avoiding the restacking caused by the Van der Waal forces, and (2) during the polymerisation, the silane groups allow the formation of covalent bonds in the interfacial surface between G and the matrix, acting as coupling agents, and creating stronger interfacial bonding between G and PMMA, which in turn improves the interfacial load transfer.

The analysis of the thermal properties demonstrated that G incorporation or silanisation did not produce significant changes in the fundamental thermal properties of bone cements. However, a minor reduction in the maximum temperature and the released heat during the polymerisation were observed as a consequence of the silanisation. This effect was attributed to the covalent reaction of silanes groups with the monomer and/or with the radical initiators during the cement polymerisation, producing slight retardation of the reaction. Finally, biocompatibility testing corroborates that the silanisation of G did not affect the biocompatibility and therefore the potential clinical application of incorporating the G_MPS into the PMMA bone cement.

## Figures and Tables

**Figure 1 nanomaterials-11-00139-f001:**
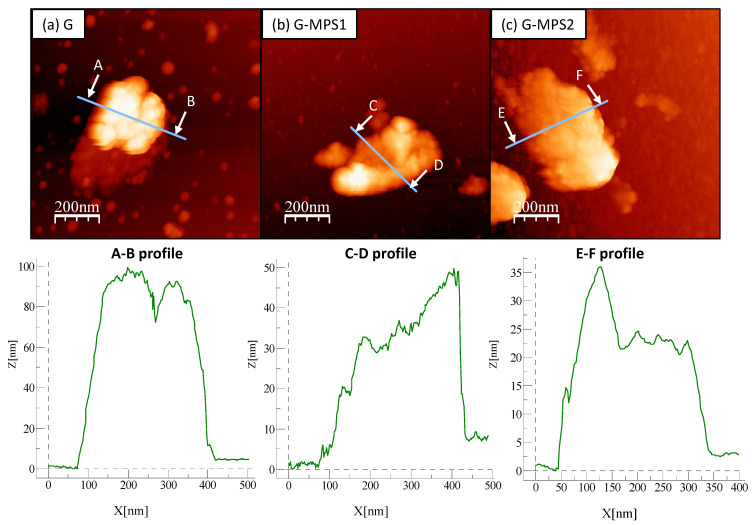
Topography atomic force microscopy (AFM) images of graphene (G) (**a**), silanised G obtained from G_Oxi1 (G_MPS1) (**b**) and from G_Oxi2 (G_MPS2) (**c**) powders deposited on mica discs, after dispersion on methyl methacrylate (MMA) liquid by ultrasonication. The bottom images correspond with the indicated z-profiles and show the thickness of the graphene clusters.

**Figure 2 nanomaterials-11-00139-f002:**
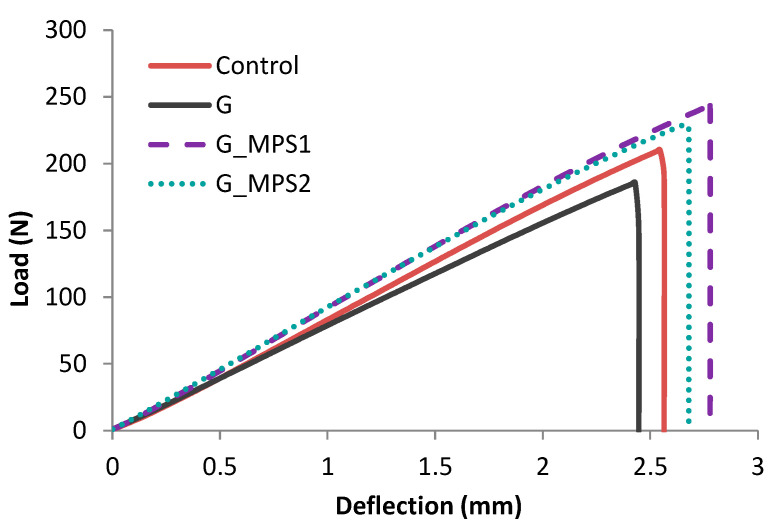
Comparison of Load (N) vs. Deflection (mm) obtained from representative bending tests for Control and bone cements with 0.1 wt.% of G, G_MPS1 and G_MPS2.

**Figure 3 nanomaterials-11-00139-f003:**
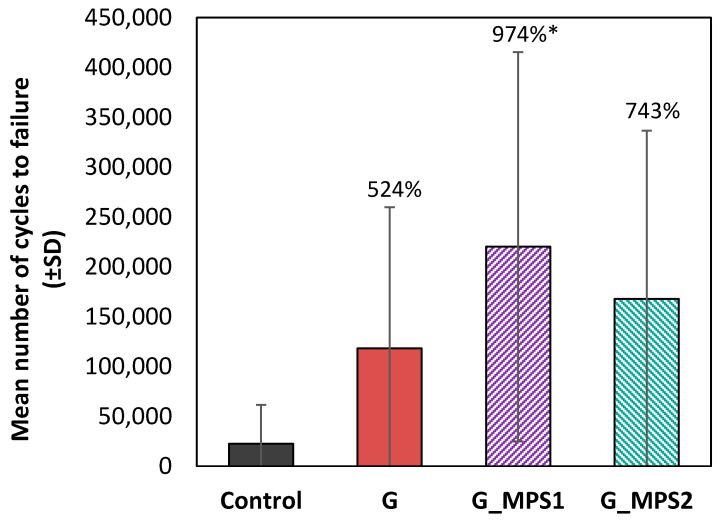
Mean number of cycles to failure (±SD) for the Control and the cements with 0.1 wt.% of G, G_MPS1 and G_MPS2. The difference of each value compared with control is given in % and significant differences (*p* < 0.05) are indicated with *.

**Figure 4 nanomaterials-11-00139-f004:**
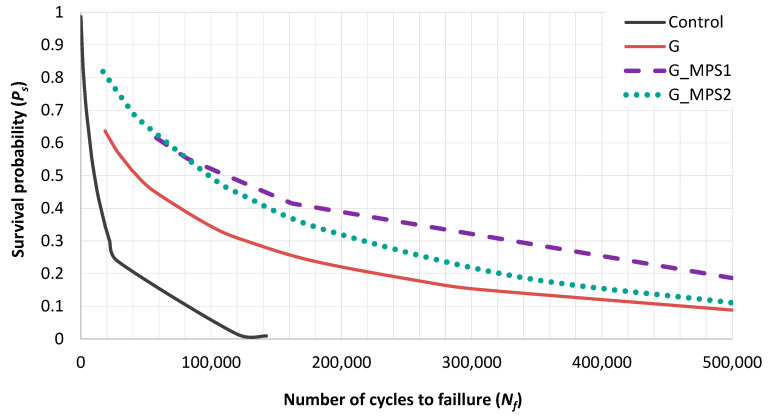
Survival probability vs. the number of cycles to failure for the Control and the cement with 0.1 wt.% of G, G_MPS1 and G_MPS2.

**Figure 5 nanomaterials-11-00139-f005:**
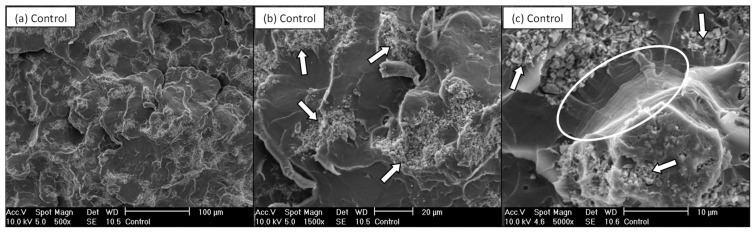
SEM micrographs of the Control bone cement fracture surfaces at 500× (**a**), 1500× (**b**) and 5000× (**c**) magnifications.

**Figure 6 nanomaterials-11-00139-f006:**
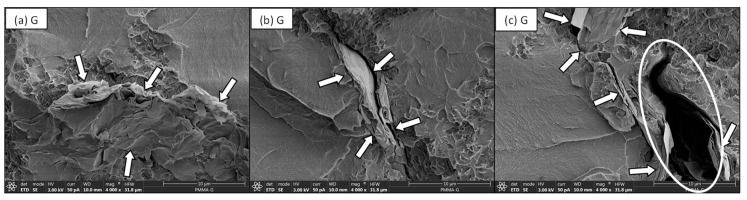
SEM micrographs of bone cement with G, at 4000× magnification (**a**–**c**) showing regions with a relatively poor interaction between the G sheets and the PMMA matrix.

**Figure 7 nanomaterials-11-00139-f007:**
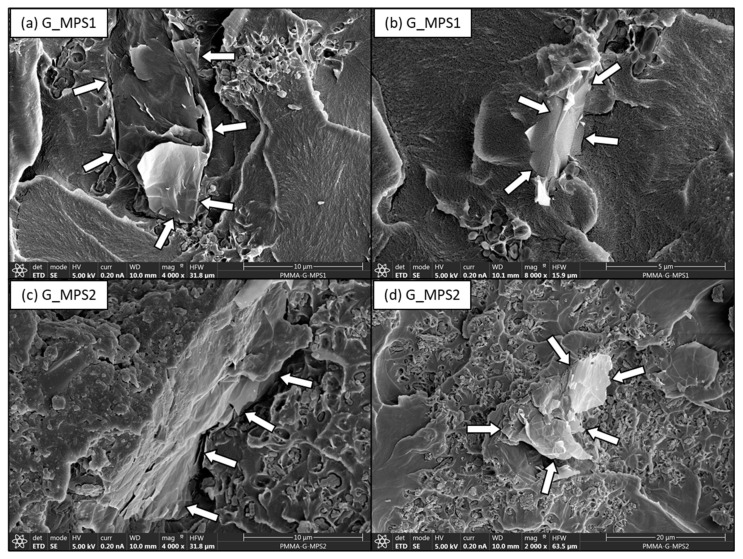
SEM micrographs of bone cement with G_MPS1 at 4000× (**a**) and 8000× (**b**) magnification; and G_MPS2 at 4000× (**c**) and 2000× (**d**) magnification, showing regions with G sheets.

**Figure 8 nanomaterials-11-00139-f008:**
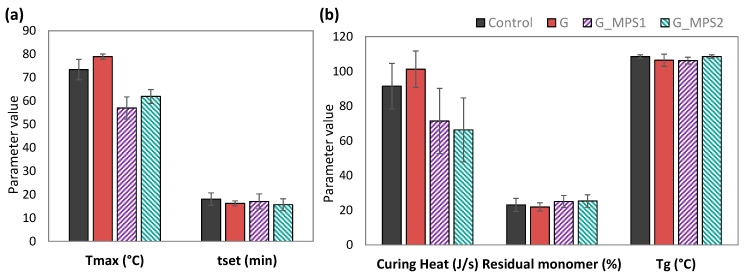
Thermal properties (mean ± SD): maximum temperature (**a**), setting time (**a**), curing heat (**b**), residual monomer (**b**) andglass transition temperature (**b**), for control and bone cements with 0.1 wt.% of G, G-MPS1 and G-MPS2.

**Table 1 nanomaterials-11-00139-t001:** Bending properties and cycles to fatigue failure (mean ± SD) for control and bone cements with 0.1 wt.% of graphene (G) and silanised G obtained from G_Oxi1 (G_MPS1) and from G_Oxi2 (G_MPS2). The difference of each value compared with control and with G of each cement is given in %.

Mechanical Property	Control	G	G_MPS1	G_MPS2
**Bending Strength (MPa)**	51.4 ± 7.8	50.4 ± 5.9	58.8 ± 9.2	58.8 ± 7.7
Difference vs. Control (%)		−1.9 (*p* = 0.997)	14.4 (*p* = 0.721)	14.4 (*p* = 0.719)
Difference vs. G (%)			16.7 (*p* = 0.323)	16.7 (*p* = 0.321)
**Bending Modulus (MPa)**	2731 ± 524	2888 ± 478	3301 ± 131	3305 ± 150
Difference vs. Control (%)		5.7 (*p* = 0.902)	20.9 (*p* = 0.999)	21.0 (*p* = 0.999)
Difference vs. G (%)			14.3(*p* = 0.642)	14.4 (*p* = 0.609)
**Deflection at Break (mm)**	2.66 ± 0.54	2.54 ± 0.28	2.96 ± 0.56	3.00 ± 0.51
Difference vs. Control (%)		−4.5 (*p* = 1.000)	11.3 (*p* = 0.442)	12.8 (*p* = 0.535)
Difference vs. G (%)			16.5 (*p* = 0.558)	18.1 (*p* = 0.315)

**Table 2 nanomaterials-11-00139-t002:** Weibull data for the different bone cements. The difference of each value relative to Control and the unsilanised G is expressed as the rate of increase (times that the reference value has increased) in all cases, except for the b parameter which is expressed in percentage (%).

Weibull Parameters	Control	G	G_MPS1	G_MPS2
**50% Probability of fracture life (*N*_50_) (cycles)**	10 × 10^3^	52.5 × 10^3^	128.8 × 10^3^	107.1 × 10^3^
Rate of increase vs. Control		(×5.3)	(×12.8)	(×10.7)
Rate of increase vs. G			(×2.5)	(×2.0)
**Weibull Minimum Fatigue Life (*N*_0_) (cycles)**	50	18.6 × 10^3^	57.9 × 10^3^	16.9 × 10^3^
Rate of increase vs. Control		(×372)	(×1158)	(×338)
Rate of increase vs. G			(×3.2)	(×0.91)
**Weibull Characteristic Life (*N_a_*) (cycles)**	17.2 × 10^3^	88.3 × 10^3^	203.1 × 10^3^	164.2 × 10^3^
Rate of increase vs. Control		(×5.1)	(×11.8)	(×9.5)
Rate of increase vs. G			(×2.3)	(×1.9)
**Slope (*b*)**	0.73	0.51	0.58	0.71
Difference vs. Control (%)		(−30)	(−21)	(−3)
Difference vs. G (%)			(13)	(39)
**Fatigue Performance Index (*I*)**	14.7 × 10^3^	63.1 × 10^3^	154.1 × 10^3^	138.1 × 10^3^
Rate of increase vs. Control		(×4.3)	(×10.5)	(×9.4)
Rate of increase vs. G			(×2.4)	(×2.2)

**Table 3 nanomaterials-11-00139-t003:** Mean numbers of MC3-T3 cells (±SD) after 72 h cultured in direct contact with the bone cement reinforced with 0.1 wt.% of each filler; the difference and the *p*-value respecting the control mean value are also indicated.

Parameter	Control	G	G_MPS1	G_MPS2
**Number of cells**	6229 ± 556	5752 ± 21	5255 ± 1475	5157 ± 922
Difference vs. Control		−7.7	−15.7	−17.2
		(*p* = 0.995)	(*p* = 0.895)	(*p* = 0.791)

## Data Availability

The data presented in this study are available on request from the corresponding author.

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
