# Peer review of "Advanced G-MPS-PMMA Bone Cements: Influence of Graphene Silanisation on Fatigue Performance, Thermal Properties and Biocompatibility"

_nanomaterials, 2021, doi:10.3390/nano11010139_

Round 1
Reviewer 1 Report
The manuscript represents interesting experimental study of graphene silanisation influence on mechanical, thermal and biological properties G‐MPS‐PMMA bone cements. The article is well written and contains new scientific results that deserve publication. But, to my mind, there are some moments that could improve the article.
- For fatigue testing authors chose «sinusoidal stress cycle with a maximum stress of 11 MPa». Why authors decided to apply sinusoidal type of stress and why maximum stress equals 11 MPa? A little explanation should be added.
- The quality of Fig.1 bottom images should be improved. Figures 2, 3, 4, 8 should be made in color to raise reader’s susceptibility.
- For bending test addition of pure graphene to cement slightly decreases its bending strength and bending modulus. Why? We don’t see this trend for fatigue testing.
- Line 478-479 should be corrected
- Addition of silanised graphene decreases biocompatibility on 15-17% (Table 3). Are there some ways to improve this indicator?
Author Response
The manuscript represents interesting experimental study of graphene silanisation influence on mechanical, thermal and biological properties G‐MPS‐PMMA bone cements. The article is well written and contains new scientific results that deserve publication. But, to my mind, there are some moments that could improve the article.
The authors thank the reviewer for the comments and valuable suggestions. We hope that the reviewer finds useful the modifications commented below and find the manuscript properly improved.
- For fatigue testing authors chose «sinusoidal stress cycle with a maximum stress of 11 MPa». Why authors decided to apply sinusoidal type of stress and why maximum stress equals 11 MPa? A little explanation should be added.
The sinusoidal stress cycle in compression-tension was chosen according the indications of the ASTM F2118 standard. The Standard recommends using a frequency less than 5 Hz to avoid frequency effects. Accordingly, a frequency of 2Hz was applied during the tests. This information has been included in the manuscript in Lines 167-171.
It has been reported that the stress levels in the bone cement mantle around a joint replacement range from 3 to 11 MPa. Consequently, a maximum stress of 11MPa has been used in order to ensure the success of the prosthesis under the most adverse conditions. A brief explanation has also been included in the manuscript in Lines 167-171.
Lines 167-171: It has been reported that the stress levels in the bone cement mantle around a joint replacement range from 3 to 11 MPa [28], consequently, a maximum stress of 11MPa has been used in order to ensure the success of the prosthesis under the most adverse conditions. A frequency of 2 Hz has been used for the test following ASTM F2118-03, that recommends not to exceed the 5 Hz to avoid frequency effects.
- The quality of Fig.1 bottom images should be improved. Figures 2, 3, 4, 8 should be made in colour to raise reader’s susceptibility.
According the reviewer suggestions, the quality of Figure 1 has been improved: bottom graphs has been upgraded. Additionally, colour has been included in Figures 2, 3, 4 and 8 to facilitate the understanding. We consider that these changes facilitate reader comprehension.
- For bending test addition of pure graphene to cement slightly decreases its bending strength and bending modulus. Why? We don’t see this trend for fatigue testing.
The slightly decrease in bending properties observed with the pure graphene is consequence of the high scattering of the experimental tests. This is supported by the ANOVA analysis that indicate a p-value of 0.997, meaning that there is practically no difference between the mean bending strength of pure graphene and the control. This has been highlighted in Line 294 and 297)
Lines 294-297: It can be observed from the bending results (Table 1), that the presence of 0.1 wt.% G did not produce significant modifications in bone cement properties according to ANOVA analysis, i.e. the bending strength decreased by 1.9% (p=0.997) and the bending modulus increased 5.7% (p=0.902) when compared with the Control.
The reason because the addition of pure G does not produces any improvement in bending tests but, nevertheless, it produces notable improvements in fatigue properties is explained in Lines 376-383. Additionally, a new explanation has been included in the manuscript in Lines 325-329 to facilitate the reader understanding.
Lines 325-329: Additionally, in the case of unsilanised G it is important to mention that the fatigue tests demonstrated notable improvements, unlike the bending tests, where no differences with respect to the control were obtained. This can be attributed to the fact that the mechanism by which the G improve the mechanical performance of composites - crack arrest and deviation - has a higher impact over the failure mechanisms produced during fatigue failure that during static tests [6].
Lines 376-383: It is important to comment that in comparison with fatigue data, the improvements obtained in bending properties were less relevant. These notable difference between the effect that G has over static mechanical properties (i.e. bending strength and bending modulus) and fatigue performance are typically reported in the literature [40,41]. This is attributed to the mechanism by which G improves the mechanical performance of materials, i.e. through crack deviation and arrest. This mechanism has a greater influence during the failure as a consequence of fatigue loading when compared to static loading, irrespective on the mode of state loading (i.e. bending, compression or tensile) [6,42].
- Line 478-479 should be corrected
It has been corrected.
- Addition of silanised graphene decreases biocompatibility on 15-17% (Table 3). Are there some ways to improve this indicator?
We are completely accord that, if only the difference with respect the control would be considering, this could be interpretated as a reduction of the biocompatibility. However, we believe that this reduction can be consequence of the scattering of the results. For that reason, the ANOVA analysis have been included, the results demonstrated that there are no significant differences in cytotoxicity between Control and silanised G, the p-values were very close to 1.000: p=0.895 for MPS1 and p=0.791 for MPS2. This study just intends to obtain a first approximation about the cytotoxicity in order to dismiss potential incompatibilities that compromise the application of silanised G, however, in view of the preliminary results, a deeper biological analysis is proposed in Lines 513 - 515.
Lines 513-515: Given the promising results obtained, future work will be completed to continue investigating the biological benefits and effects of the G_MPS_PMMA bone cements and also the potential drawbacks with a deeper biological study.
Reviewer 2 Report
The Review Article nanomaterials-1054008 titled "Advanced G‐MPS‐PMMA bone cements ‐ Influence of graphene silanization on fatigue performance, thermal properties and biocompatibility" submitted to Nanomaterials by Paz E., Dunne N., and co-authors proposes to investigate the impact that the incorporation of graphene powder has to the mechanical performance of PMMA-based bone cement. The presented work is the prosecution of a work described by some of the authors in 2019 [Ref.20]. In particular, the authors tried here to improve graphene dispersibility within the matrix and improve their reciprocal adhesion following a graphene-silanization strategy. An accurate study of the resulting material mechanical, thermal and biocompatible properties was performed. The most relevant aspect of the study was obtained for the mechanical properties under fatigue loading. The incorporation of modified graphene significantly increased the material's fatigue performance compared to unsilanized graphene or the cement without graphene.
The manuscript is globally well written and organized. The evaluation of pristine and graphene-modified cements' mechanical properties was well done from both experimental and analysis/statistical points of view.
In the Reviewer opinion, the research is mainly a material-engineering work where the nanomaterial/nanotechnological part is missing or not sufficiently addressed. Consequently, in the present form, the manuscript appears more suitable for publication in a material science or engineering journals than in a more nanotechnological one as nanomaterials.
Specifically, the Reviewer has some critical points to rise:
(1) The nanotechnological aspects present in this work are: the use of a manometric material (graphene), and the attempt to exploit nanotechnological investigation approaches (e.g., AFM). Although the graphene used in the paper is a commercial one, any characterization of the material before its use was done: what about the flakes' effective lateral dimensions? Has a Raman analysis been performed to point out the effective number of layers of the graphene composing the powder? All the material properties come from the product data-sheet and do not respect the state of the material at the end of the different processing-steps it undergoes during material production (e.g., suspension, oxygenation, blending, drying, silanization, etc. etc.). Investigating the "transformation" of the graphene at every step from the as-received powder to the final composite material is a mandatory study that has to be included in the work if a nanomaterial journal will be its target.
(2) Why graphene has to be used and not just graphite? What is the advantage to have only a few layers of carbon? Is this aspect so critical for the improvements observed in the graphite-blended cement? Is there a biocompatible reason for this choice? In this latter case, a more in-depth discussion has to be included in the manuscript's introduction.
(3) On page 3 at rows 112-115 authors claim that silanization is possible taking advantage of oxygenated groups present in Oxide Graphene or inducing oxidation in pristine graphene. In both cases, the precise determination of the number of reactive groups (strongly influencing the degree of silanization) present in the graphene was not performed. This is another critical point that has to be addressed by authors. For example, how many hydroxy functional groups are present per graphene flake?
(4) The Reviewer do not understand the purpose of the AFM investigation here performed. The analysis is very superficial (page 4 row 187, a tip radius of 10 µm is an ingenuous refuse) and does not increase our understanding of the material. In particular, the investigation focuses on massive graphene flakes agglomerates and not on the estimation of flakes alteration during processing. Just some questions:
How many "blobs" were analyzed? Do two-layers graphene flakes compose them? What about the surface density of these "blobs" for every condition? How the line profiles were chosen for every "blob" and why a more statistically representative height-distribution analysis was not preferred instead?
Are the small spots observable in the background of figure 1a single graphene flakes? Why authors focus on the aggregates and not on these features? What about the lateral dimensions and height of the pristine flakes? What about these values after oxidation and/or silanization?
The Reviewer would like to suggest including or considering TEM characterization to highlight graphene flakes modifications.
The AFM investigation here proposed is not giving any information about graphene exfoliation but just about graphene propensity to form aggregates. Unfortunately, also in this regard, the presented analysis is very superficial and not conclusive.
(5) Page 5 row 237, a space has to be added between the number and the unit (100 µL)
Concluding, the work has to be fortified from the nanotechnological point of view including, at least, a real (nano)material characterization and a study about the impact of graphene flakes' lateral dimensions, number of layers, and degree of oxidation/functionalization on the level of improvement in cement mechanical properties. The study has to be done with well in mind the concept of flakes agglomeration highlighting the strategies that could be exploited to govern graphene dispersion. The use of graphene agglomerates, in fact, is not providing the sufficient level of control required for a reproducible and safe use of this nanomaterial for cement modification.
Author Response
Please see the attachment with the response to the reviewer comments.

Reviewer 3 Report
The manuscript, which is part of a continuous work from this research group, reports on the formation of functionalized graphene silanized PMMA bone cements. The effect in fatigue strength, thermal properties as well as biocompatibility are described. Overall, the manuscript presented the data and explanation in a systematical manner,
The authors should include one of their reported articles and clearly describe on the difference from this manuscript. Part of the data for cell viability test is taken from this article
https://www.mdpi.com/1996-1944/12/19/3146/htm
Author Response
The authors would like to thank the reviewer the suggestions to improve the quality of the manuscript. The differences between the previous work and this one has been clarified in the manuscript (Lines 498-501). The value of the number of cells (table 3) of the unsilanised G has been taken from previous work and the corresponding reference has been include in the text. This data corresponds with a previous study that evaluated the cytotoxicity of bone cements reinforced with graphene (G) and graphene oxide (GO). On the contrary, the present manuscript is focused on the silanisation of Graphene with the aim to improve the previous reported results obtained with the unsilanised G (all data related silanised G are completely new). With the aim to demonstrate the benefits of the proposed silanised method the authors have considered appropriate to include the data of the Control and unsilanised G just with comparative purposes.
Lines 498-501: The cell viability of unsilanised G bone cements was reported in previous works where their biocompatibility was demonstrated [55], in the current work the cell viability of silanised G was evaluated in order to ensure that the silanisation did not affect the biocompatibility. The data of unsilanised G was also included in Table 3 with comparative purposes.
Round 2
Reviewer 1 Report
This manuscript can be published.
Author Response
The authors would like to thank the editor for their time and help.
Reviewer 2 Report
The revised manuscript nanomaterials-1054008 titled "Advanced G‐MPS‐PMMA bone cements ‐ Influence of graphene silanization on fatigue performance, thermal properties and biocompatibility" submitted to Nanomaterials by Paz E., Dunne N., and co-authors proposes to investigate the impact that the incorporation of graphene powder has to the mechanical performance of PMMA-based bone cement.
As previously stated by the Reviewer the manuscript is globally well written and organized. The evaluation of pristine and graphene-modified cements' mechanical properties was well done from both experimental and analysis/statistical points of view. Anyhow, in the Reviewer opinion, the research is mainly a material-engineering work where the nanomaterial/nanotechnological part is just marginal. The revised version of the manuscript does not address this aspect. Consequently, the manuscript remains more suitable for a material science or engineering journals with just a minimal relevance for the nanotechnological research field.
Author Response
Dear Reviewer,
We would like to thank you for the time taken to read our manuscript and for your valuable comments. We consider that all of them are really interesting and we regret that we can not ammended the manuscript including a more nanotechnological point of view.
As we commented previouslly, the complete characterization of silanised graphene (FTIR, SEM, XPS, TGA, etc ...) has already been published in a previous work that is duly referenced in the current manuscript, therefore, we can not include that information on this manuscript. The present manuscript is the continuation of this previous article, and the objective is not the characterization of the silanised G, but to demonstrate that this silanised G can improve the dispersability and provide significant advantages in a biomedical application such as the improvement of the performance of bone cements. We believe that this point of view may also be of great interest to Nanomaterials readers, especially in the case of this special issue, focused on the biomedical application of graphene-based nanomaterials.
Anyhow, we really appreciate all your comments, and in the case that the paper will be rejected, we will take into account your advise and we will send it to a materials-engineering journal.
Best wishes and Happy new year,
Eva